# Enhancing Microalgae Content in Biocomposites through a Mechanical Grinding Method

**DOI:** 10.3390/polym15234557

**Published:** 2023-11-28

**Authors:** Minju Kim, Gyu Min Kim, Won-Seok Chang, Young-Kee Kim

**Affiliations:** 1Department of Chemical Engineering, Research Center of Chemical Technology, Hankyong National University, Anseong 17579, Gyeonggi-do, Republic of Korea; minju@hknu.ac.kr; 2Frontier R&D Institute, Korea District Heating Corp., Yongin-si 17099, Gyeonggi-do, Republic of Korea; wschang@kdhc.co.kr

**Keywords:** microalgae, biocomposites, ball milling, tensile strength

## Abstract

Microalgae-based biocomposites are gaining traction as ecofriendly and cost-effective alternatives to conventional petroleum-based plastics. However, achieving a homogeneous dispersion of microalgae within a biocomposite matrix remains a challenge. In this study, we investigated the effect of the size of dried microalgae (*Chlorella* sp.) on the quality of biocomposites. Ball milling, a mechanical grinding process, was used to control the size of the pretreated dried microalgae. Our results demonstrate that the microalgae size strongly depends on the total weight of the stainless-steel balls, rather than the number of balls used in the milling process. Poly(ethylene-vinyl acetate) (EVA), with functional groups resembling those of *Chlorella* sp., was incorporated into the ball-milled microalgae to produce homogeneous biocomposites. Smaller *Chlorella* sp. particles improved the ratio of microalgae and the mechanical properties of the biocomposites. Dried *Chlorella* sp. particles up to 161.43 μm, which were 72.84% smaller than the untreated microalgae, were obtained after 6 h of ball milling using 3/8-inch balls. This enabled the production of biocomposites with 60 wt.% microalgae and 61.02% of the tensile strength of pure EVA, comparable to traditional polymers. Our findings suggest that controlling the microalgae size through ball milling can improve the quality of microalgae-based biocomposites.

## 1. Introduction

The use of conventional petroleum-based plastics has a significant effect on the environment and has caused global pollution. Over 380 million tons of plastics are produced annually worldwide, resulting in pollutants that pose a threat to the natural environment [1,2]. The prolonged decomposition period of petroleum-based plastics presents a hazard to living organisms, as microplastics can be absorbed into the body through the ingestion of crustaceans and fish [3,4]. The adverse effects of plastics on human health have been extensively documented, with impacts on physiological functions including the digestive, nervous, and respiratory systems [5,6,7].

Microalgae-based biocomposites (MBBs) are promising, eco-friendly, and biodegradable alternatives to traditional petroleum-based plastics. MBBs are complex blends of microalgae and general-purpose polymers that can reduce environmental pollution, meet the growing demand for plastics, and decrease reliance on fossil fuels [8,9,10,11]. MBBs also absorb CO_2_, which contributes to a reduction in atmospheric carbon. However, the process of mixing microalgae and petroleum-based polymers to form MBBs faces technical challenges due to the physical and chemical differences between the materials, which hinder the formation of a homogeneous mixture and adversely affect the final product quality [8].

Researchers have attempted to solve these problems by drying and modifying the surfaces of the microalgae and petroleum-based polymers [12,13,14,15,16]. Although these pretreatments improve the miscibility of the blend to some extent, obstacles remain in achieving uniformity in the size of the dried microalgae particles, which is crucial for ensuring the proper mixing of the two materials [17,18,19,20].

Research on biomass-based biocomposite materials is a highly significant field. In seven previous research studies, a biocomposite was synthesized using PVA, incorporating a mixture of 20% LEA (microalgae) and 12% PD, resulting in a total composition of 68% PVA [21]. In previous research, it was also established that the maximum synthesis ratio achievable was 20%. It was observed that mixing beyond this threshold, specifically exceeding 30%, posed limitations in the composite formation process.

This research explores avenues for the effective utilization of biomass. The focal point of our research revolved around determining the maximum extent to which microalgae content could be increased and how the physicochemical properties and characteristics change concerning the mixing ratio. This emphasis on exploring the upper limits of microalgae content and dissecting the alterations in physicochemical properties based on mixing ratios set our study distinctly apart from prior research efforts. Furthermore, it played a pivotal role in identifying the optimal blending ratio, driving pioneering innovations in the realm of biocomposite material research.

To address these issues, we propose an efficient method for grinding microalgae using ball milling. Despite its widespread use in industries including ceramics, minerals, materials science, and pharmaceuticals, the ball milling of microalgae has not been extensively studied [22,23]. In this study, we investigated the feasibility of ball milling *Chlorella* sp., a type of microalgae, and found that it resulted in a significantly reduced particle size. We discovered that the degree of particle size reduction in *Chlorella* sp. was strongly dependent on the total weight of the stainless-steel balls used in the ball-milling process, rather than the size of the balls. After considering cost–performance, we determined that an optimum 3/8-inch ball radius produced a *Chlorella* sp. powder size of 161.43 μm after 6 h of treatment.

To further investigate the effect of particle size on the compatibility and properties of bioplastics, we fabricated MBBs consisting of ground *Chlorella* sp. and poly(ethylene-vinyl acetate) (EVA). As expected, the smaller *Chlorella* sp. showed superior miscibility with EVA, allowing for an increased ratio of microalgae in the MBBs. More than 50% of *Chlorella* sp. with a size of 161.43 μm was incorporated with EVA; the microalgae content of *Chlorella* sp. was limited to 30% when not treated with ball milling. We performed a comprehensive analysis of the physical and chemical characteristics of the MBBs, including measurements of mechanical properties such as tensile strength, Young’s modulus, elongation at break, and thermal stability using thermogravimetric analysis (TGA). Our results indicated that the incorporation of ground *Chlorella* sp. into MBBs improved their mechanical properties. The addition of ground *Chlorella* sp. led to an increase in tensile strength and elongation at break (EB).

In this study, the primary objective is to enhance the performance of biocomposites and optimize the manufacturing process. Taking inspiration from previous studies using PVC, which demonstrated that microalgae have a similar structure to poly(ethylene-vinyl acetate) (EVA), we utilized EVA in this study. Our goals were to optimize the process of blending microalgae and EVA, as well as to reduce the size of microalgae particles through ball milling. The central objective of our research was to determine the optimal mixing ratio of microalgae and EVA.

## 2. Experiment

### 2.1. Materials

The microalgal biomass of *Chlorella* sp. HS2 was sourced from cultured and desiccated microalgae at Korea University in Seongbuk-gu, Republic of Korea. Microalgal cells were cultivated using a modified TAP-C medium, wherein the absence of an organic carbon source was compensated for through the inclusion of a bicarbonate buffer as a substitute for Tris-base. The photobioreactor was supplied with LNG-fired flue gas comprising 3–6% CO_2_, 11.99 ± 0.73% O_2_, 21.72 ± 3.72 ppm NO_x_, and 1.43 ± 4.03 ppm CO, in addition to water vapor and particulate matter. Gas infusion occurred at the lowermost section of the photobioreactor and was maintained at a constant flow rate of 0.1 gas volume per vessel volume per minute (vvm). After cultivation, the HS2 cells were harvested using an industrial centrifuge (A/T-075, Hanil SCI-MED, Daejeon city, Republic of Korea). After harvesting, the cells were subjected to controlled heat–spray-drying. It is noteworthy that ball milling was exclusively applied to microalgae in this study.

In this study, microalgae-based composite materials were synthesized via integration with poly(ethylene-vinyl acetate) (EVA). The EVA (melt index = 57 g/10 min, T_m_ = 110–120 °C, vinyl acetate content = 40 wt.%) was procured from Sigma (Sigma-Aldrich, St. Louis, MO, USA) and used as the substratum for the manufacturing protocol.

### 2.2. Controlled Ball-Milling Process Parameters

Throughout the ball-milling process, the milling parameters were carefully controlled. To achieve a total ball weight of 50 g, the number of balls was adjusted accordingly: 140 EA for 1/8-inch, 47 EA for 2/8-inch, 14 EA for 3/8-inch, and 6 EA for 4/8-inch balls. The microalgae were mixed with the balls in a ratio of 5:2 by weight to ensure a balanced composition. The ball-milling speed was set to 200 rpm [24], and the process was conducted at room temperature. The milling container has a volume of 125 mL, and we used 50 g of balls and 20 g of the material during the milling process. The small volume of the milling jar and the size of the balls make it challenging to provide a detailed explanation of the heating effect. The experimental results showed a minimal temperature difference of only 2–3 degrees between the initial and final temperatures, indicating a negligible impact of potential heating during the 12 h milling process.

Ball milling was performed in containers with a volume of 125 mL to ensure consistent experimental conditions. 

### 2.3. Preparation of Microalgae-Based Composite and Specimen

Prior to mixing, the microalgae powder had an initial moisture content of 12% and was stored in vacuum conditions. The ball milling was performed using ball sizes of 1/8 in, 2/8 in, 3/8 in, and 4/8 in, with equal weights. The ball milling was performed using a desktop ball mill (LK Lab Korea, Namyangju, Republic of Korea) operating at a rotor speed of 200 rpm. The milling times were 0.5 h, 1 h, 1.5 h, 2 h, 2.5 h, 3 h, 6 h, 9 h, and 12 h. The temperature during the mixing phase was maintained at 180 °C. All composite materials were shaped into disk configurations (diameter 12.5 mm, thickness 2 mm) at their designated positions using a specimen-molding machine (QM130N, QMESYS, Uiwang, Republic of Korea). The mixing temperature was maintained for 12 min to facilitate the evaluation of rheological and mechanical properties.

### 2.4. Characterization

#### 2.4.1. Particle Size and Distribution

The initial particle size of the microalgae was determined via a particle size analysis using a Mastersizer 3000 (Malvern Panalytical, Malvern, UK). This analytical apparatus enabled the quantification of essential particle size parameters, including the mean diameter, size distribution, and particle count. The outcomes of this analysis significantly contributed to our understanding of the physical attributes of microalgal particles, improving our overall understanding of their intrinsic characteristics.

#### 2.4.2. Morphological Observation

Scanning electron microscopy (SEM) analysis was conducted for the morphological characterization of the isolated microalgal strains using a desktop SEM (TM-1000, Hitachi, Tokyo, Japan). For the SEM observations, the composite samples were sectioned to expose their internal surfaces. Flattening was performed to expose the internal surfaces of the samples, and post-staining was performed to highlight specific analysis areas. The coated samples were analyzed at 500× and 1000× magnification. The SEM analysis was valuable for assessing the mixed structure, morphology, and distribution of the microalgal samples. It also allowed for a detailed examination of the internal structures of the cells.

#### 2.4.3. Fourier Transform Infrared Spectroscopy (FT-IR)

The transmittance FT-IR spectra of the composite materials and microalgae samples were measured across the spectral range of 4000–600 cm^−1^, with 40 scans per measurement. The FT-IR spectra were obtained using a V-650 FT-IR spectrometer (Jasco, Tokyo, Japan). For the FT-IR analysis, the specimens were disc-shaped (diameter: 25 mm, thickness: 1 mm) for both the pure polymers and the composite materials. The powdered form of the microalgae was used directly. Approximately 1–2 mg of the powdered sample was positioned in the FT-IR spectroscopic testing area for analysis.

#### 2.4.4. Mechanical Tensile Test

The mechanical tensile properties of the pure polymers and composite materials were systematically assessed using a universal tensile machine (UTM) (QM100S, QMESYS, Uiwang, Republic of Korea). Each polymer and composite specimen (disc-shaped structures, diameter: 25 mm, thickness: 1 mm) underwent a minimum of three individual tests. The data were averaged to derive representative tensile characteristics, including the yield strength, tensile strength, and elongation at break. The applied strain rate for the deformation process was standardized at 50 mm/min.

##### Tensile Strength

Tensile strength measures the maximum stress that a material can withstand under tension. It is determined by recording the maximum stress during a tensile test. This parameter indicates how physically strong the material is and its ability to resist breaking.

##### Elongation at Break

Elongation at break quantifies how much a material can be stretched or elongated before it breaks. Expressed as a percentage, it measures the material’s stretchability and ductility. It is determined by measuring the deformation after the material breaks.

##### Modulus of Elasticity

The modulus of elasticity, also known as Young’s modulus, characterizes a material’s ability to resist deformation when subjected to an external force. This value describes how the material responds to stretching or compression within a certain range. It is a critical parameter for evaluating a material’s strength and deformation characteristics.

#### 2.4.5. Thermal Analysis

A thermal analysis of the specimens was conducted using a thermal analyzer in different air and nitrogen atmospheres. A thermal gravimetric analysis (TGA) of the microalgae, polymers, and composite materials was performed using a TGA instrument (TG209, NETZSCH-Gerätebau GmbH, Bavaria, Munich, Germany) in a controlled nitrogen (N_2_) gas atmosphere. The analysis encompassed gradual temperature elevation from room temperature to 600 °C, with a fixed heating rate of 30 °C/min and a flow rate of 20 mL/min. The assessment yielded valuable data regarding the mass loss as a function of temperature, as captured in the differential thermal analysis profile. The sample weights used in the analyses were 10–15 mg.

#### 2.4.6. X-ray Diffraction Analysis

To determine the crystal structures, powder X-ray diffraction (Rigaku, Tokyo, Japan) experiments were conducted between 2θ = 10° and 90° using a diffractometer with a Cu Kα radiation source (40 kV and 40 mA), with a Linxeye 1-D detector, a divergence angle of 0.2°, a step size of 0.02°, and an acquisition time of 5 s per step. 

## 3. Results and Discussion

The microalgae were the *Chlorella* sp. HS2 variety. The microalgae underwent a pretreatment process involving freeze-drying, followed by temporary mortar grinding. The initial size of the prepared microalgae was estimated to be approximately 600 µm (Appendix A Appendix A), as measured through the particle analysis. However, owing to its considerable particle size, the effective miscibility of *Chlorella* sp. HS2 with petroleum-based polymers for bioplastic formulation was hindered. To address this issue, a grinding process with ball milling was used for unprocessed microalgae (Figure 1a) (Appendix A Appendix A). Ball milling relies on mechanical grinding; critical parameters including the rotation speed, ball quantity, and total ball weight collectively influence the final size of *Chlorella* sp. HS2 (detailed information is presented in the section on controlled ball-milling process parameters). Additional details summarizing the overall process are presented in (Appendix A Appendix A). The diameter of the balls is a variable that directly impacts factors such as the size of the crushed product and the efficiency of ball milling [25,26]. The following formula (Equation (1)) was used to calculate the optimal ball diameter:(1)$$S_{i}\propto 1/d^{No},N_{o}\approx 1.0$$

Equation (1) expresses the relationship between the ball diameter (d) and the grinding rate (Si). In the experimental milling, $N_{o}$ is 0, meaning there is no impact due to the ball diameter. Considering the representative unit volume of the mill, as the diameter of the ball decreases, the collision velocity between the balls and the material increases. This is because the number of balls within the mill increases by 1/d^3^. Therefore, as the ball diameter decreases, the collision velocity between the balls and the material increases, resulting in higher grinding rates [24,27]. In our actual study, using a 125 mL container as the standard, ball milling was performed with ball diameters of 1/8, 2/8, 3/8, and 4/8 inches. During this process, rotation speeds of 100 rpm, 200 rpm, and 300 rpm were tested. The findings indicated that a rotation speed of 200 rpm was the most suitable under these conditions. This choice of rotation speed was particularly effective, taking into account the container size.

Figure 1b shows the influence of the stainless-steel ball size (1/8 in, 2/8 in, 3/8 in, 4/8 in) on the particle size of *Chlorella* sp. HS2, while maintaining a constant total ball weight (50 g). The results in Figure 1b underscore the efficacy of the ball-milling process in reducing the microalgae particle dimensions. A treatment time of 6 h led to a reduction in particle size to approximately 30% of the initial microalgae size. Further ball milling can lead to a reduction in particle size. Milling parameters such as the ball size, ball-to-powder ratio, and milling time are well known to be highly effective in decreasing the particle size after the process [24,28]. For instance, in the study titled the “Influence of milling parameters on particle size of ulexite material”, it was found that these parameters play a crucial role in determining the final particle size of the material. However, there comes a point in the milling process where further ball milling may yield only marginal reductions in particle size. This phenomenon is attributed to a variety of factors, including practical limitations, the initial size of the particles, and the saturation point of the milling process. In our study, we observed similar behavior, where further ball milling did not result in substantial reductions in particle size. This can be attributed to the fact that, after a certain point, the impact of milling parameters reaches a plateau, and additional milling does not lead to significant improvements in particle size reduction. Smaller ball diameters generally result in a higher milling speed, which leads to further size reduction in most cases. However, it is worth noting that with 4/8-inch balls, there was suboptimal grinding performance, and the size of microalgae particles remained relatively stable over time with a fixed total ball weight. From an economic perspective, the use of 3/8-inch balls stands out as optimal, considering both the number of balls and the constant total weight.

The particle size distribution of microalgae with 3/8-inch ball milling is displayed in Figure 1c, with an average particle size of 197.99 µm after 6 h of mechanical treatment. Microalgae subjected to ball milling for 9 h and 12 h exhibited average sizes of 153.38 µm and 134.99 µm, respectively.

To validate the efficacy of the finely ground microalgae, MBBs were fabricated through an amalgamation of finely ground microalgae with EVA, an established petroleum-based polymer (Figure 2a). The predetermined microalgae-to-EVA ratio in the composite MBBs was maintained at 3:7 (by weight). The pristine EVA material exhibited an excellent tensile strength of 4 MPa. The tensile strength decreased to 0.8 Mpa with the incorporation of microalgae (20% of the tensile strength of unadulterated EVA). However, a trend of increasing tensile strength was observed with increasing the microalgae ball-milling time. MBBs with microalgae treated for 6 h via ball milling displayed a tensile strength of 3 MPa (75% of pristine EVA tensile strength). The EB and modulus of elasticity also increased with prolonged ball milling of the microalgae (Figure 2b,c), underscoring the pivotal role of the microalgae’s initial size in governing the tangible attributes of the MBBs.

To further investigate the intricate interplay between the microalgae particle size and miscibility in the EVA matrix, SEM images of MBBs containing microalgae were obtained, each bearing imprints of distinct ball-milling times (Figure 2d–g). The images revealed that microalgal particles, when left untreated, were inclined to aggregate within the EVA matrix (Figure 2d), a phenomenon seemingly rooted in the nonuniformity of size coupled with the prevalence of larger particles. However, as the ball-milling time was extended to 1.5 h (correlating with an average particle size of 359.58 µm), aggregation was mitigated, attributed to the reduced microalgal particle dimensions and an improved distribution of microalgae within the matrix (Figure 2e). MBBs treated with microalgae for 3 h showed improvements in the incorporation of microalgae in the EVA, as shown in Figure 2f (fine microalgae fragments within EVA matrix). A ball-milling time of 6 h, culminating in an average microalgae particle size of 150 µm, yielded a well-dispersed integration of microalgae particles in the EVA matrix (Figure 2g), indicating homogeneous physical properties across discrete MBBs regions, ultimately translating into increased tensile strength. To enhance clarity, the caption now includes a description elucidating the figures. The red circle denotes the reduction in microalgae size observed in the bio-composite material as the ball milling time increased. Specifically, Image (d) depicts the microstructure at 0 h of ball milling, while (e) corresponds to 1.5 h, (f) to 3 h, and (g) to 6 h.

Consistent with prior observations, particle sizes remain a pivotal determinant in shaping the ultimate properties of the MBBs, as underscored by Figure 3a, which demonstrates that the tensile strength values of MBBs incorporating microalgae, with different ball sizes and a 6 h ball-milling time, are almost indistinguishable for 1/8-inch, 2/8-inch, and 3/8-inch ball sizes. This uniformity can be attributed to the consistent particle sizes, irrespective of the ball size, with a 6 h ball-milling time (Figure 3b). The tensile strength of MBBs with microalgae, produced using a 4/8-inch ball size, was lower. This decrease in strength was primarily due to the larger particle sizes, even when ball milling was conducted for more than 6 h (as depicted in Figure 1b). This clarifies that larger particle sizes led to the reduction in tensile strength in the MBBs with microalgae, despite extended ball milling [24].

Considering the implications, we find 3/8-inch balls in the ball-milling process for microalgae grinding to be optimal, aligning with economic considerations and superior performance metrics of MBBs. With this rationale, microalgae subjected to ball milling with 3/8-inch balls constituted the principal focus of this investigation, unless otherwise stated.

Figure 4a shows the physical attributes of MBBs with different ball-milling treatment times and weight ratios of microalgae/EVA (spanning 3:7, 4:6, 5:5, 6:4, and 7:3). In accordance with our previous observations, extended microalgae treatment times increased the tensile strength across the spectrum of composite formulations, irrespective of the microalgae content within the MBBs. This was attributed to the reduced dimensions of microalgae particles stemming from prolonged ball-milling operations.

In our study, the behavior of MBBs with varying microalgae contents was explored, drawing on insights from prior research. This investigation revealed that increasing the microalgae content from 30% to 60% led to a modest attenuation in tensile strength. This phenomenon was attributed to the decrease in the EVA content, a component inherently characterized by substantial tensile strength. These findings align with earlier studies, which investigated the tensile behavior of composites with different reinforcing materials. As observed in our study, the tensile strength exhibited a significant decrease upon introducing 30% microalgae by weight. However, further reductions in tensile strength were relatively marginal as the microalgae content increased by up to 60% in the MBBs. Specifically, for MBBs containing microalgae treated for 6 h using ball milling, those composed of 60% microalgae demonstrated a tensile strength of 2.4 MPa, retaining approximately 80% of the tensile strength observed in MBBs containing 30% microalgae (Figure 4a). This behavior contrasts with some earlier research, such as [29], which investigated EVA composites incorporating different fillers like clay, talc, and natural fibers. In those cases, variations in tensile strength were more pronounced at different filler concentrations. Our study underscores the unique compatibility between microalgae, specifically HS2, and the EVA matrix. This compatibility allows for the effective dispersal of microalgae within the EVA, resulting in improved tensile performance. These findings further highlight that microalgae, even without surface modifications, can serve as a promising reinforcing filler material for biocomposite applications. However, the increase in microalgal content exerted a significant influence on the EB, a pivotal parameter that reflects material ductility. In contrast to pristine EVA material, which exhibited an exceptionally high EB value of 1130%, MBBs with untreated microalgae exhibited EB values of 100% (for 30% microalgae content) and 15% (for 60% microalgae content). Even with microalgae undergoing 6 h of ball milling, resulting in a relatively robust EB value of 550% for a weight ratio of microalgae (30%) to EVA (70%), the increase in the microalgae content to 60% led to a pronounced decline in the EB value (40%), underscoring the inherent limitations associated with increased microalgae content, necessitating favorable elasticity as a fundamental material attribute. 

Conversely, an increase in the microalgae content caused a notable increase in the modulus of elasticity (MOE). Whilst pure EVA displayed an MOE of 1.04 MPa, MBBs with escalated microalgae content, in their untreated state, exhibited MOE values of 2.69 MPa for 30% microalgae content and 6.34 MPa for 60% microalgae content. After 6 h of ball milling, MBBs with 30% microalgae content displayed an MOE of 1.49 MPa, whereas those with 60% microalgae content exhibited an MOE of 5.28 MPa, implying that the MOE is inversely proportional to the EB.

The FT-IR spectroscopic analysis revealed prominent peaks corresponding to nitrogen (N)-related functional groups at a wavenumber of 2300 cm^−1^ in MBBs containing microalgae; such peaks were conspicuously absent in the spectrum of pure EVA. This discrepancy can be attributed to the abundance of proteins within the microalgal matrix, as shown in Figure 4d. To further investigate the thermal behavior of MBBs with different microalgae contents, a TGA was performed, as illustrated in Figure 4e,f. For pristine EVA, degradation initiates near 320 °C, with an inflection point at 370 °C on the weight–temperature curve, culminating in complete volatilization at 490 °C (Figure 4f).

In contrast, the TGA curve of the microalgae did not exhibit a distinct weight loss (Figure 4e) or no significant inflection point (Figure 4e), indicating the presence of intricate components encompassing proteins, lipids, carbohydrates, and other impurities within the microalgae composition. In Figure 4e,f, there are two distinct transitions observed at around 370 °C and 490 °C. These transitions follow a pattern similar to the TGA curve of EVA. The first transition, occurring around 370 °C, is attributed to the breakdown of the acetate groups in EVA, while the second transition at around 470 °C is due to the degradation of the PE backbone [30,31,32,33,34,35,36,37]. The absence of complete weight loss in MBBs containing microalgae is pronounced; a plateau region extending beyond 500 °C signifies the persistence of residual materials as it does not reach 0%. Consequently, MBBs with higher proportions of microalgae demonstrated escalated weight percentages of residues at temperatures surpassing 500 °C, with values of 12.6%, 18.4%, 25.8%, and 32.1% for microalgae loadings of 30%, 40%, 50%, and 60%, respectively.

SEM images were used to visualize MBBs containing varying microalgal quantities, as depicted in Figure 4. These images showcase the uniform integration of finely powdered microalgae, subjected to 6 h of ball milling, across MBBs with 30% (Figure 5a), 40% (Figure 5b), 50% (Figure 5c), and 60% (Figure 5d) microalgae contents. The blue circle was utilized to highlight the uniformly mixed microalgae, whereas the red circle emphasized areas displaying irregularly shaped microalgae particles and agglomeration, attributed to the influence of van der Waals forces of attraction [38].

The XRD spectrum demonstrated that, in the case of MBBs, the unique peaks of EVA are significantly broad due to the inherent amorphous nature of the polymer (Figure 6). The peaks observed in the XRD are primarily attributed to the amorphous nature of the materials. Particularly in the case of the powdered samples, the results appeared relatively lower compared to EVA or MBBs. In the case of MBBs, it was observed that as the particle size decreased through the ball milling process, the intensity of these peaks increased. Moreover, based on previous research, ball milling was expected to transform the crystal structure into an amorphous one with increased milling time. Consequently, this experiment also yielded results consistent with an amorphous structure as the ball-milling process continued, in line with the findings of prior studies. However, for pure EVA, although the peak intensity was relatively higher compared to other MBBs or microalgae peaks, it is still considered to originate from amorphous characteristics. As most of the peaks are attributed to amorphous features, it is challenging to ascertain a specific orientation, and the differences are assumed to stem from the broad categorization of peaks.

## 4. Conclusions

This study explores the viability of a mechanical grinding approach to process *Chlorella* microalgae to produce finely powdered microalgae suitable for subsequent use. The investigation validated the feasibility of producing finely powdered microalgae through a ball-milling process using 3/8-inch balls. Although ball-milling times exceeding 12 h resulted in a notable reduction in the microalgae particle size (approximately 10 µm), it was established that a treatment period of 6 h was sufficient for the optimal incorporation of microalgae into the EVA matrix.

In this study took a different approach, focusing on a blend of EVA which shares structural similarities with the microalgae *Chlorella* sp. MBBs were successfully fabricated by blending finely powdered microalgae with EVA. This shift in focus allowed us to achieve substantial enhancements in both mechanical and chemical properties. Most notably, unlike prior research, we achieved a remarkable feat by increasing the microalgae content to a maximum of 60%. The mechanical performance evaluation of the resulting MBBs indicated a tensile strength of approximately 2.5 MPa, a noteworthy achievement (approximately 62.5% of the tensile strength of pristine EVA). Given that the proportion of microalgae surpassed that of conventional petroleum-based polymers, the produced MBBs possessed attributes of environmentally sustainable bioplastics. This elevated mixing ratio set our research apart from previous studies and played a pivotal role in determining the optimal blend, driving innovative progress in the field of biocomposite material research. Successfully, MBBs were created through the blending of finely powdered microalgae with EVA. These blends accommodated microalgae contents of up to 60 wt.%. The mechanical performance evaluation of the resulting MBBs revealed an impressive tensile strength of approximately 2.5 MPa, marking a significant achievement (equivalent to about 62.5% of the tensile strength of pure EVA). It is acknowledged that the performance of MBBs is inferior to pristine EVA. It is crucial to highlight the substantial influence of the ball-milling process. For instance, MBBs containing 60% untreated microalgae exhibited a tensile strength of only 0.7 MPa, approximately one-fourth that of MBBs with 60% treated microalgae. This increase in the mixing ratio set our research apart from previous studies. It played a crucial role in finding the optimal mixing ratio and spearheading innovative advancements in the field of biocomposite material research. This observation underscores an alternative approach for the effective preconditioning of microalgae, engendering high-performance MBBs that align with economic viability. The combination of HS2 (microalgae) and EVA in this biocomposite material holds significant environmental value and offers substantial potential as a futuristic material resource.

## Figures and Tables

**Figure 1 polymers-15-04557-f001:**
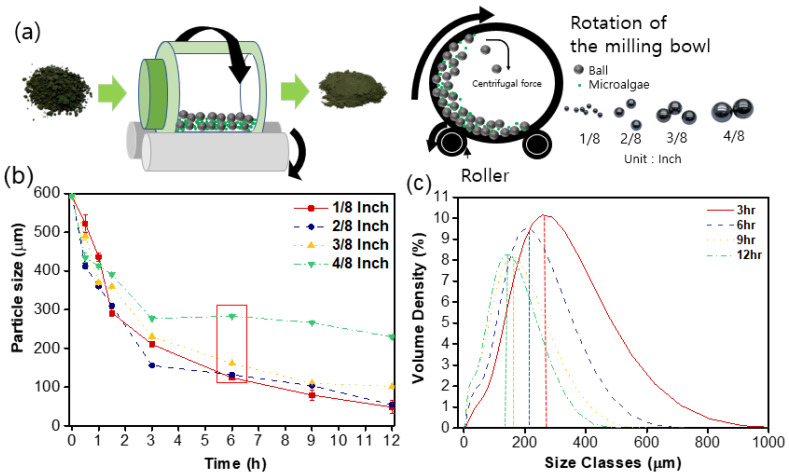
(**a**) Process diagram of ball-milling experiment; (**b**) comparison of particle size changes for all ball sizes with different ball-milling process times; (**c**) particle size distribution for 3/8-inch ball-milling process time.

**Figure 2 polymers-15-04557-f002:**
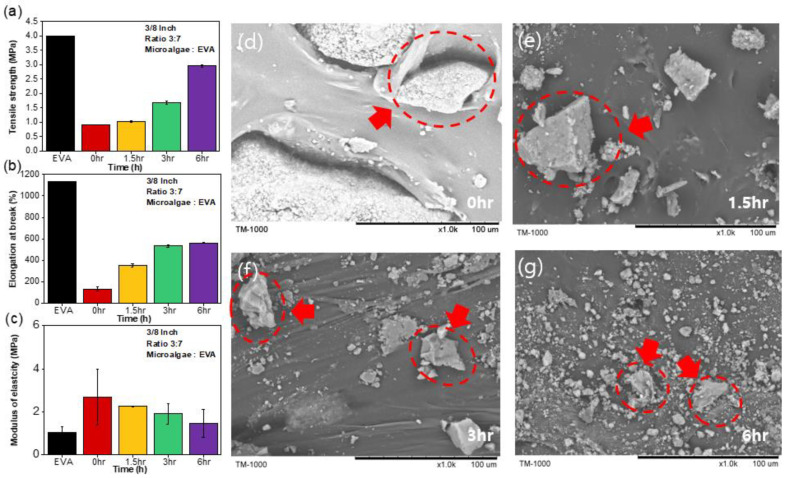
Comparison of tensile strength (**a**); elongation at break (**b**); Young’s modulus (**c**); and particle size (**d**–**g**) with respect to ball-milling time for microalgae/EVA ratio of 3:7 and 3/8-inch balls.

**Figure 3 polymers-15-04557-f003:**
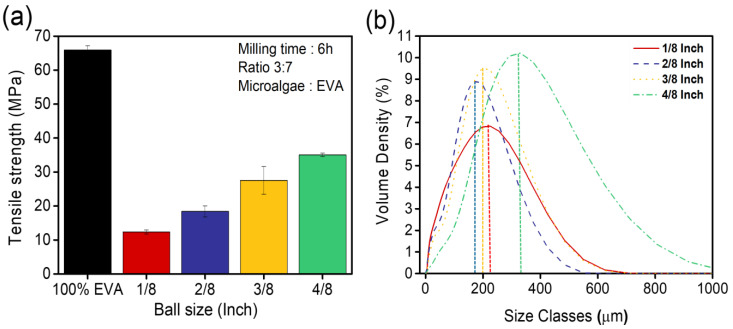
(**a**) Comparison of property changes according to ball size/EVA composite ratio of 3:7 and 6 h ball-milling time; (**b**) particle size distribution according to ball size with 6 h ball-milling time.

**Figure 4 polymers-15-04557-f004:**
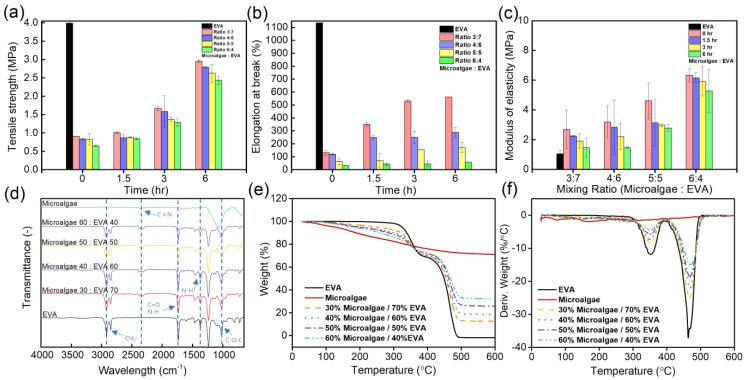
(**a**) Tensile strength, (**b**) elongation at break, and (**c**) Young’s modulus variations over time with 3/8-inch balls; (**d**) FT-IR analysis based on microalgae: EVA composite ratios (6 h ball-milling with 3/8-inch balls); (**e**) TGA analysis for HS2/EVA composite ratios and (**f**) corresponding weight derivative analyses (6 h ball-milling with 3/8-inch balls).

**Figure 5 polymers-15-04557-f005:**
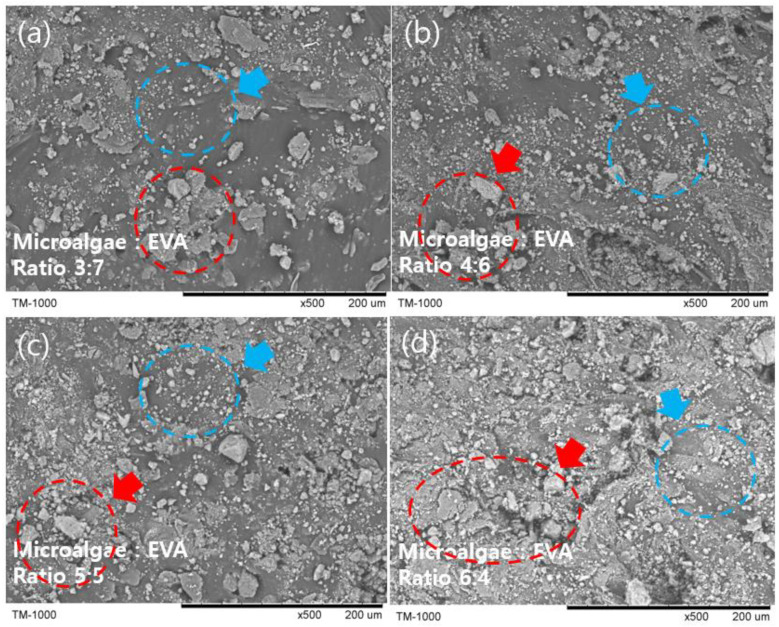
SEM analysis of microalgae/EVA composite ratios with 3/8-inch ball milling for 6 h: (**a**) 3:7 ratio; (**b**) 4:6 ratio; (**c**) 5:5 ratio; (**d**) 6:4 ratio.

**Figure 6 polymers-15-04557-f006:**
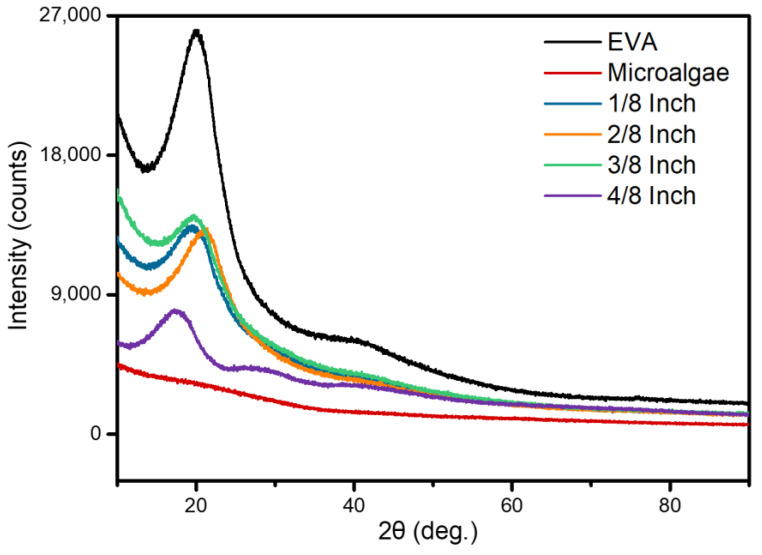
XRD analysis of microalgae/EVA composite ratios with 3:7 ratio for 6 h of ball milling.

## Data Availability

Data are contained within the article and Appendix A.

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
