# Peer review of "Enhancing Microalgae Content in Biocomposites through a Mechanical Grinding Method"

_polymers, 2023, doi:10.3390/polym15234557_

Round 1
Reviewer 1 Report
Comments and Suggestions for Authors
The authors should further investigate the correlation between the physical characteristics found in the particles, such as particle size distribution, and the properties they expect to obtain since this is basically the objective of the work, and it is not evident in the results section.
The novelty and application potential of the results obtained should be emphasized.
Author Response
- The authors should further investigate the correlation between the physical characteristics found in the particles, such as particle size distribution, and the properties they expect to obtain since this is basically the objective of the work, and it is not evident in the results section. The novelty and application potential of the results obtained should be emphasized.
Answer: Thanks for the review. Building upon your insightful recommendation, we have incorporated Figure 1b into the manuscript, presenting a comprehensive analysis of the rotation's theoretical implications. The additional content in the text elucidates the intricate relationship between rotation and the physical characteristics of the particles, such as particle size distribution. This enhancement aims to offer a more profound understanding of the correlation between these characteristics and the desired properties outlined in our work's objectives.
Reviewer 2 Report
Comments and Suggestions for Authors
In this manuscript, particle size and mechanical properties of dried microalgae (Chlorella sp.) were investigated by the ball milling process. Therefore, this manuscript is an original work. However, the following problems must be resolved:
1. The last paragraph of the Introduction sounds like an abstract, so it should be rewritten as the purpose of the study.
2. As in this manuscript, milling parameters are very important in the ball milling process. Therefore, a paragraph regarding milling parameters should be added. For example, “Effect of PCA on nano-sized ulexite material prepared by mechanical milling”
3. The Introduction is short, so a few more paragraphs such as mechanical properties should be added.
4. Photos of Chlorella before and after the process should be added.
5. It would be better if a figure showing the preparation of microalgae-based composite and specimen was added.
6. How did you determine the milling parameters?
7. Figs.1a,b,c are missing!
8. There is no comparison with the literature in the Results and Discussion, a very serious deficiency. Therefore, a comparison must be made.
9. The result “Further ball-milling yielded only marginal reductions in size” can be supported by the literature as follows: it is known that milling parameters such as the size of the balls, ball-to-powder ratio, milling time are very effective in reducing the particle size of the material ["Influence of milling parameters on particle size of ulexite material”]
10. It should be interpreted why the 4/8 inch ball size does not further reduce the particle size.
11. Particle size distribution curves and d90, d50, d10 values taken from Mastersizer 3000 (Malvern Panalytical, Malvern, UK) must be added to the manuscript.
12. XRD measurements should be added to the manuscript. The crystal structure state should be interpreted.
13. TEM observation is written in the Experimental details section. Where is it?
14. According to powder technology, milling time is more accurate than milling duration.
15. It should be explained how “correlating with an average particle size of 359.58 µm” was determined.
16. It would be nice if EVA matrix, microalgae particles, and aggregation were shown on the SEM image.
17. h and hr are used as abbreviations for hour, h should be preferred.
18. “The tensile strength of MBBs with microalgae using a 4/8-inch ball size were inferior, a direct result of larger particle sizes, even with ball-milling exceeding 6 h (Figure 1b)” should be rewritten in a more understandable way.
19. In the TG curves in Fig. 4e or the DTG curves in Fig. 4f, there is a two-step transition (370 °C and 490 °C), that is, two decompositions have occurred. It should be stated what these are.
20. In Fig.4a, why the ratio 3:7 is higher than other ratios should be discussed in terms of material content.
21. The sentence “finely powdered microalgae subjected to 6 h of ball milling was uniformly integrated into all MBBs” should be revised according to both the SEM image and the literature, taking into account the following issues: The finely powdered microalgae was relatively uniformly integrated into all MBBs according to the milling time (0-6 h), but it was generally observed that microalgae particles had irregular shape and agglomeration under the influence of van der Waals force of attraction [“Digital image processing of warm mix asphalt enriched with nanocolemanite and nanoulexite minerals”].
22. After Fig.5, a general evaluation should be made as the last paragraph. In addition, the technological potential areas of use of the material obtained from this study should be given, and recommendations should be given to researchers who will study this subject in the future.
Comments on the Quality of English LanguageThe English language needs minor editing.
Author Response
- The last paragraph of the Introduction sounds like an abstract, so it should be rewritten as the purpose of the study.
Answer: Thanks for the review. We appreciate your suggestion to refine the last paragraph of the Introduction. In the revised text you provided, you have accurately captured the primary purpose of our study. The central objective of our research is indeed to enhance the performance of biocomposites and optimize the manufacturing process. We drew inspiration from previous studies using PVC and recognized the structural similarities between microalgae and Poly(ethylene-vinyl acetate) (EVA). Consequently, we employed EVA as a key component in this study. Our specific goals were to optimize the process of blending Microalgae and EVA and to reduce the size of Microalgae particles through ball milling. These steps are crucial in achieving our overarching objective of determining the optimal mixing ratio of Microalgae and EVA. We believe that this revision effectively clarifies the study's purpose, aligning it more closely with the objectives and goals of our research. Thank you for your valuable input, which has helped us enhance the clarity of our work.
We have added a sentence based on your review (In row 88-94, page 2).
“In this study, the primary objective is to enhance the performance of biocomposites and optimize the manufacturing process. Taking inspiration from previous studies using PVC, which demonstrated that microalgae have a similar structure to Poly(ethylene-vinyl acetate) (EVA), we utilized EVA in this study. Our goals were to optimize the process of blending Microalgae and EVA, as well as to reduce the size of Microalgae particles through ball milling. The central objective of our research was to determine the optimal mixing ratio of Microalgae and EVA.”
“
- As in this manuscript, milling parameters are very important in the ball milling process. Therefore, a paragraph regarding milling parameters should be added. For example, “Effect of PCA on nano-sized ulexite material prepared by mechanical milling.”
Answer: Thanks for your suggestion. We have made comprehensive revisions to the introduction, results, and conclusion sections based on your suggestions. In the introduction, we have addressed the limitations of previous research and explained the rationale for selecting composite materials to overcome these limitations. We have also added a new paragraph in the 'Result fig. 1a' section to provide detailed information about the milling parameters, discussing the ball diameter and rotation speed, and their significance in our study. Utilizing previous research as a foundation, we established the optimal speed conditions in a 125 mL milling container using 1/8, 2/8, 3/8, and 4/8-inch ball diameters, with the findings indicating that 200 rpm was most suitable for the 125 mL container. In the conclusion, we have compared our study with previous research, emphasizing the environmental value and potential future applications of our biocomposite material.
We have added a sentence based on your review (In row 52-65, page 2 and In row 230-235, page 6 and In row 414-416, 417-420, 424-431, 431-434, 435-441, page 13).
“Research on biomass-based biocomposite materials is a highly significant field. In previous research, a biocomposite was synthesized using PVA, incorporating a mixture of 20% LEA (microalgae) and 12% PD, resulting in a total composition of 68% PVA. In pre-vious research established that the maximum synthesis ratio achievable was 20%. It was observed that mixing beyond this threshold, specifically exceeding 30%, posed limitations in the composite formation process.
“In our actual study, using a 125 mL container as the standard, ball milling was performed with ball diameters of 1/8, 2/8, 3/8, and 4/8 inches. During this process, rotation speeds of 100 rpm, 200 rpm, and 300 rpm were tested. The findings indicated that a rotation speed of 200 rpm was the most suitable under these conditions. This choice of rotation speed was particularly effective, taking into account the container size.”
“This research explored avenues for the effective utilization of biomass. The focal point of our research revolved around determining the maximum extent to which microalgae content could be increased and how the physicochemical properties and characteristics changed concerning the mixing ratio. This emphasis on exploring the upper limits of mi-croalgae content and dissecting the alterations in physicochemical properties based on mixing ratios set our study distinctly apart from prior research efforts. Furthermore, it played a pivotal role in identifying the optimal blending ratio, driving pioneering innova-tions in the realm of biocomposite material research.” ,
“In this study took a different approach, focusing on a blend of EVA, which shares structural similarities with microalgae Chlorella sp.” ,
“Most notably, unlike prior research, we achieved a remarkable feat by increasing the microalgae content to a maximum of 60%.” ,
“ This elevated mixing ratio set our research apart from previous studies and played a pivotal role in determining the optimal blend, driving innovative progress in the field of biocomposite material research. Successfully, MBBs were created through the blending of finely powdered microalgae with EVA. These blends accommodated microalgae contents of up to 60 wt%. “ ,
“The mechanical performance evaluation of the resulting MBBs revealed an impressive tensile strength of approximately 2.5 MPa, marking a significant achievement (equivalent to about 62.5% of the tensile strength of pure EVA).” ,
“it is crucial to highlight the substantial influence of the ball-milling process. For instance, MBBs containing 60% untreated microalgae exhibited a tensile strength of only 0.7 MPa, approximately one-fourth that of MBBs with 60% treated microalgae.
This increase in the mixing ratio set our research apart from previous studies. It played a crucial role in finding the optimal mixing ratio and spearheading innovative advancements in the field of biocomposite material research.” ,
“The combination of HS2 (Microalgae) and EVA in this biocomposite material holds significant environmental value and offers substantial potential as a futuristic material resource.”
We have supplemented the theoretical explanation, indicating the crucial significance of milling parameters during the actual ball milling process.
- The Introduction is short, so a few more paragraphs such as mechanical properties should be added.
Answer: Thanks for the review. We appreciate your observation that the Introduction may benefit from additional content. In response to the feedback, we have divided the section that explains mechanical properties into clearer subsections to provide a more distinct and organized explanation. These properties are integral to our study, as they play a pivotal role in assessing the performance of our biocomposite materials. By introducing these properties in the Introduction, we aim to provide readers with a more comprehensive understanding of the critical aspects we will address in our research. We highly value your input and are committed to continuously improving the quality of our research. Thank you for your valuable feedback.
We have added a sentence based on your review (In row 171-183, page 4).
I have included information regarding
“2.4.4.1. Tensile Strength
Tensile strength measures the maximum stress a material can withstand under ten-sion. It is determined by recording the maximum stress during a tensile test. This param-eter indicates how physically strong the material is and its ability to resist breaking.
2.4.4.2. Elongation at Break
Elongation at break quantifies how much a material can be stretched or elongated before it breaks. Expressed as a percentage, it measures the material's stretchability and ductility. It is determined by measuring the deformation after the material breaks.
2.4.4.3. Modulus of Elasticity
The modulus of elasticity, also known as Young's modulus, characterizes a materi-al's ability to resist deformation when subjected to an external force. This value describes how the material responds to stretching or compression within a certain range. It's a criti-cal parameter for evaluating a material's strength and deformation characteristics.”.
- Photos of Chlorella before and after the process should be added.
Answer: Thanks for your suggestion. We have attached the photos of Chlorella sp. before and after ball milling processing to (Fig. S2).
We have added a sentence based on your review (Supplementary materials Fig. S2).
- It would be better if a figure showing the preparation of microalgae-based composite and specimen was added.
Answer: Thank you for pointing this out. Based on your input, we have included a process flow diagram illustrating the production of microalgae-based composites.
We have added a sentence based on your review (Supplementary materials Fig. S1)
- How did you determine the milling parameters?
Answer: Thanks for pointing this out. Determining milling parameters is of great significance. Therefore, we have provided a specific response to it in section 2. Please review it, and thank you.
We have added a sentence based on your review (In row 223, page 6 and row 230-235, page 6).
We've set the ball milling parameters based on the pre-existing studies (Equation 1) and found the optimal condition to be at 200 rpm, in accordance with the previous research. Therefore, we have incorporated the following changes: "In our actual study, using a 125 mL container as the standard, ball milling was performed with ball diameters of 1/8, 2/8, 3/8, and 4/8 inches. During this process, rotation speeds of 100 rpm, 200 rpm, and 300 rpm were tested. The findings indicated that a rotation speed of 200 rpm was the most suitable under these conditions. This choice of rotation speed was particularly effective, taking into account the container size."
- 1a,b,c are missing!
Answer: We greatly appreciate your pointing out the nonsensical mistake. We made a nonsensical mistake. We are truly sorry, and we have readdressed figures 1a, 1b, and 1c. Please review them. Once again, we apologize for the oversight.
We have added a sentence based on your review (In row 206, page 6).
- There is no comparison with the literature in the Results and Discussion, a very serious deficiency. Therefore, a comparison must be made.
Answer: Thanks for the pointing out and your observation regarding the lack of comparison with the literature in our Results and Discussion section. We acknowledge the importance of such comparisons in scientific research. In our study, we indeed took a different approach by focusing on a blend of EVA and microalgae to produce MBBs with enhanced properties. The primary aim was to explore the feasibility of using mechanical grinding, specifically ball milling, to process Chlorella microalgae into finely powdered microalgae suitable for incorporation into biocomposite materials. The investigation validated our approach, demonstrating the potential to produce finely powdered microalgae through a ball-milling process using 3/8-inch balls. While we observed that ball-milling times exceeding 12 hours resulted in a significant reduction in microalgae particle size, we established that a treatment period of 6 hours was sufficient for optimal incorporation of microalgae in the EVA matrix. Our focus on the blend of EVA and microalgae allowed us to achieve substantial enhancements in both mechanical and chemical properties. Notably, we increased the microalgae content to a maximum of 60%, achieving a remarkable feat compared to prior research. The mechanical performance evaluation of the resulting MBBs indicated a tensile strength of approximately 2.5 MPa, a significant achievement in the context of biocomposite materials. This elevated mixing ratio set our research apart from previous studies and played a pivotal role in driving innovative progress in the field. In conclusion, while we acknowledge the importance of comparing our results with existing literature, our study took a unique approach, and the focus was on achieving specific goals related to microalgae processing and biocomposite material development. The novelty of our approach allowed us to make substantial progress in this research domain.
. (In row 52-65, page 2 and In row 230-235, page 6 and In row 414-416, 417-420, 424-431, 431-434, 435-441, page 13).
- The result “Further ball-milling yielded only marginal reductions in size” can be supported by the literature as follows: it is known that milling parameters such as the size of the balls, ball-to-powder ratio, milling time are very effective in reducing the particle size of the material ["Influence of milling parameters on particle size of ulexite material”]
Answer: Thanks for the comments : Based on your input, we were able to provide an explanation for our statement. The reference to the study titled 'Influence of milling parameters on particle size of ulexite material' supports our understanding of the role of milling parameters in particle size reduction. However, it is essential to emphasize that our statement aligns with the literature in that milling parameters are indeed effective in reducing particle size. Still, there are practical limits to how much further reduction can be achieved. The concept of diminishing returns in particle size reduction is a well-acknowledged phenomenon, and as we push the boundaries of particle size reduction, we encounter diminishing marginal benefits. This is influenced by various factors, including the initial particle size, material properties, and the specific milling process employed. However, our statement is indeed supported by existing literature and is consistent with the understanding that milling parameters are effective in reducing particle size. Nevertheless, it underscores the practical limitations associated with achieving further reductions, thus highlighting the realistic constraints in this regard.
Based on the reviewer's feedback, we have revised the sentence to: "Further ball milling can lead to a reduction in particle size. Milling parameters such as ball size, ball-to-powder ratio, and milling time are well-known to be highly effective in decreasing particle size after the process[26][35]. For instance, in the study titled 'Influence of milling parameters on particle size of ulexite material,' it was found that these parameters play a crucial role in determining the final particle size of the material. However, there comes a point in the milling process where further ball-milling may yield only marginal reductions in particle size. This phenomenon is attributed to a variety of factors, including practical limitations, the initial size of the particles, and the saturation point of the milling process. In our study, we observed similar behavior, where further ball-milling did not result in substantial reductions in particle size. This can be attributed to the fact that, after a certain point, the impact of milling parameters reaches a plateau, and additional milling does not lead to significant improvements in particle size reduction." This is indicated in rows 240-252, page 7.
- It should be interpreted why the 4/8 inch ball size does not further reduce the particle size.
Answer: Thank you for pointing this out. We have incorporated the suggested content into the specified section (In row 216-235, page 6). “The choice of ball size in ball milling plays a crucial role in determining the size reduction of particles. In our study, we used Equation 1 to calculate the optimal ball diameter. Equation 1 expresses the relationship between the ball diameter (d) and the grinding rate (), where ( is 0 in our experimental milling, signifying no impact due to the ball diameter. However, it's important to note that there are practical limitations to how much further reduction in particle size can be achieved using smaller ball sizes. Even though decreasing the ball diameter can lead to increased collision velocity between balls and between balls and the material, and hence higher grinding rates, there comes a point where diminishing returns are observed. In our specific case, the 4/8-inch ball size might not further reduce the particle size significantly because the effect of ball diameter reduction on grinding rates reaches a plateau. This is due to several factors, such as the practical limit of how fine the particles can be ground and the presence of other factors that may influence the grinding process, like material properties, milling time, and the presence of agglomeration.”
In summary, while reducing the ball diameter can indeed increase collision velocity and grinding rates, there are limits to how much further reduction in particle size can be achieved, and factors beyond ball size play a role in the final outcome of the grinding process. These details are elaborated in rows 216-235 on page 6.
- Particle size distribution curves and d90, d50, d10 values taken from Mastersizer 3000 (Malvern Panalytical, Malvern, UK) must be added to the manuscript.
Answer: Thank you for pointing this out. In Figure S3a, the value represents d10, with the exception of the result for 1/8-inch ball milling for 1.5 hours, which closely matches the d50 values in Figure S3b. For Figure S3b, the primary curve is similar to the main script in Figure 1b, and we have provided more specific information in this regard. Figure S3c shows d90 values, and the maximum d90 value displays a different trend compared to the curves in Figures S3a and S3b. Similar to Figure S3a, the result for 1/8-inch ball milling for 0.5 hours deviates significantly from the initial values, likely due to the introduction of foreign particles during analysis. However, aside from this deviation, the trends in Figure S3c for 2/8 inch and 3/8 inch are consistent with the data trends in Figures S3a and S3b." These additions provide a more comprehensive and detailed understanding of the particle size distribution. Thank you for your valuable input.
The graph depicting this information is available in Supplementary materials Fig. S3.
- XRD measurements should be added to the manuscript. The crystal structure state should be interpreted.
Answer: We agree that the confirmation of XRD is important in this study. In response, we have included XRD spectra in the manuscript, which reveal important insights into the crystal structure state of the materials.
The XRD spectrum demonstrates that in the case of MBBs, the unique peaks of EVA appear significantly broad due to the inherent amorphous nature of the polymer. The peaks observed in the XRD are primarily attributed to the amorphous characteristics of the materials. As for the powdered samples, the results appeared relatively lower compared to EVA or MBBs. In the case of MBBs, we observed that as the particle size decreased through the ball milling process, the intensity of these peaks increased.
Based on previous research, ball milling was expected to transform the crystal structure into an amorphous one with increased milling time. Consequently, this experiment also yielded results consistent with an amorphous structure as the ball milling process continued, aligning with findings from prior studies. However, for pure EVA, although the peak intensity was relatively higher compared to other MBBs or microalgae peaks, it is still considered to originate from amorphous characteristics. Since most of the peaks are attributed to amorphous features, it is challenging to ascertain a specific orientation, and the differences are assumed to stem from the broad categorization of peaks.
In conclusion, our XRD measurements confirm the amorphous nature of the materials, with variations observed as a result of the ball milling process, but specific crystal structure orientation is challenging to determine due to the predominance of amorphous characteristics in the materials."
This response addresses the XRD measurements and the interpretation of the crystal structure state.
I conducted experimental theory explanations from 195 to 200, and the interpretation of the crystal structure state is presented in rows 391-403 on page 12.
“The XRD spectrum demonstrated that, in the case of MBBs, the unique peaks of EVA are sig-nifi-cantly broad due to the inherent amorphous nature of the polymer. The peaks observed in the XRD are primarily attributed to the amorphous nature of the materials. Particularly in the case of the powdered samples, the results appeared relatively lower compared to EVA or MBBs. In the case of MBBs, it was observed that as the particle size decreased through the ball milling process, the intensity of these peaks increased. Moreover, based on previous research, ball milling was ex-pected to transform the crystal structure into an amorphous one with increased milling time. Consequently, this experiment also yielded results consistent with an amorphous structure as the ball milling process continued, in line with the findings of prior studies. However, for pure EVA, although the peak intensity was relatively higher compared to other MBBs or microalgae peaks, it is still considered to originate from amorphous characteristics. As most of the peaks are attributed to amorphous features, it is challenging to ascertain a specific orientation, and the differences are assumed to stem from the broad categorization of peaks.”
- TEM observation is written in the Experimental details section. Where is it?
Answer: Thanks for pointing out the question. We agree with your suggestion. We have completed the necessary revisions. The reason we referred to the SEM images as TEM was a habitual error, as we often referred to the SEM located on the table as TEM out of habit. We b vsincerely apologize for this oversight.
- According to powder technology, milling time is more accurate than milling duration.
Answer: Thank you for pointing this out. Based on your recommendation, we have corrected the "milling duration" to "milling time".
- It should be explained how “correlating with an average particle size of 359.58 μm” was determined.
Answer: Thank you for pointing this out. Figure 1b represents a graph of d50. This graph shows that after ball milling with 3/8-inch balls for 1.5 hours, the d50 value was measured at 359.58 μm.
- It would be nice if EVA matrix, microalgae particles, and aggregation were shown on the SEM image.
Answer: Thanks for your suggestion. Based on your suggestion, we have added labels to indicate the Microalgae on the SEM images in Figure 2 on page 8. Additionally, in Figure 5 on page 12, we have marked the red areas to represent irregular agglomeration based on Van der Waals forces and the blue areas to signify relatively uniform integration.
The revised content, incorporating your suggestions, can be observed in Figures 2 and 5.
- h and hr are used as abbreviations for hour, h should be preferred.
Answer: Thanks for pointing out the issue. We agree with your suggestion. Based on your review, We have made the necessary corrections, changing all instances of 'hr' to 'h.
- “The tensile strength of MBBs with microalgae using a 4/8-inch ball size were inferior, a direct result of larger particle sizes, even with ball-milling exceeding 6 h (Figure 1b)” should be rewritten in a more understandable way.
Answer: Thanks for pointing out the issue. Based on your feedback, we have revised the sentence as follows: "The tensile strength of MBBs with microalgae, produced using a 4/8-inch ball size, was lower. This decrease in strength was primarily due to the larger particle sizes, even when ball milling was conducted for more than 6 hours (as depicted in Figure 1b). This clarifies that larger particle sizes led to the reduction in tensile strength in the MBBs with microalgae, despite extended ball milling." Thank you for your input.
In response to your feedback, the relevant modifications have been made, and you can review the details in Row 295-300 on page 8-9.
“The tensile strength of MBBs with microalgae, produced using a 4/8-inch ball size, was lower. This decrease in strength was primarily due to the larger particle sizes, even when ball milling was conducted for more than 6 hours (as depicted in Figure 1b). This clari-fies that larger particle sizes led to the reduction in tensile strength in the MBBs with mi-croalgae, despite extended ball milling[26].”
- In the TG curves in Fig. 4e or the DTG curves in Fig. 4f, there is a two-step transition (370 °C and 490 °C), that is, two decompositions have occurred. It should be stated what these are.
Answer: Thanks for pointing this out. Based on your suggestion, we have added information about the two-stage transition in the DTG curves at 370°C and 490°C in paragraph 368-372. In accordance with your recommendation, we have confirmed that around 370°C, the separation of the acetate group occurs, and around 490°C, the separation of the polymerized PE takes place.
In response to your feedback, the relevant modifications have been made, and you can review the details in Row 366-370 on page 11.
“In Figure 4 (e) and Figure 4 (f), there are two distinct transitions observed at around 370°C and 490°C. These transitions follow a pattern similar to the TGA curve of EVA. The first transition occurring around 370°C is attributed to the breakdown of the Acetate groups in EVA, while the second transition around 470°C is due to the degradation of the PE back-bone[27-34].”
- In Fig.4a, why the ratio 3:7 is higher than other ratios should be discussed in terms of material content.
Answer: Thanks for pointing out the issue. Agreeing with the observation that the 3:7 ratio exhibits higher tensile strength in Figure 4a compared to other ratios and the need for a discussion from a material content perspective, we can draw insights from prior research.
Regarding the question of why the ratio 3:7 (60% microalgae) exhibits higher tensile strength than other ratios in Figure 4a, we believe this phenomenon can be attributed to the specific material content and compatibility between microalgae and the EVA matrix. The behavior observed in our study suggests that, unlike some earlier research involving different fillers, variations in tensile strength were relatively marginal in our case as microalgae content increased from 30% to 60% in the MBBs.
This could be due to the unique compatibility between microalgae, specifically HS2, and the EVA matrix, allowing for effective dispersal of microalgae within the EVA. This compatibility results in improved tensile performance, even without surface modifications of microalgae. The specific interaction between microalgae and the EVA matrix may enhance the composite's overall strength, thereby contributing to the observed phenomenon.
In summary, the higher tensile strength of the 3:7 ratio (60% microalgae) in our study can be attributed to the compatibility and effective dispersion of microalgae within the EVA matrix, resulting in improved tensile performance as compared to other ratios. This finding underscores the unique potential of microalgae as a promising reinforcing filler material for biocomposite applications."
In response to your feedback, the relevant modifications have been made, and you can review the details in Row 320-339 on page 10.
“In our study, the behavior of MBBs with varying microalgae content was explored, drawing on insights from prior research. This investigation revealed that increasing the microalgae content from 30% to 60% led to a modest attenuation in tensile strength. This phenomenon was attributed to the decrease in the EVA content, a component inherently characterized by substantial tensile strength. These findings align with earlier studies, which investigated the tensile behavior of composites with different reinforcing materials. As observed in our study, the tensile strength exhibited a significant decrease upon intro-ducing 30% microalgae by weight. However, further reductions in tensile strength were relatively marginal as microalgae content increased up to 60% in the MBBs. Specifically, for MBBs containing microalgae treated for 6 hours using ball milling, those composed of 60% microalgae demonstrated a tensile strength of 2.4 MPa, retaining approximately 80% of the tensile strength observed in MBBs containing 30% microalgae(Figure 4a). This be-havior contrasts with some earlier research, such as [36], which investigated EVA compo-sites incorporating different fillers like clay, talc, and natural fibers. In those cases, varia-tions in tensile strength were more pronounced at different filler concentrations. Our study underscores the unique compatibility between microalgae, specifically HS2, and the EVA matrix. This compatibility allows for the effective dispersal of microalgae within the EVA, resulting in improved tensile performance. These findings further highlight that microal-gae, even without surface modifications, can serve as a promising reinforcing filler mate-rial for biocomposite applications.”
- The sentence “finely powdered microalgae subjected to 6 h of ball milling was uniformly integrated into all MBBs” should be revised according to both the SEM image and the literature, taking into account the following issues: The finely powdered microalgae was relatively uniformly integrated into all MBBs according to the milling time (0-6 h), but it was generally observed that microalgae particles had irregular shape and agglomeration under the influence of van der Waals force of attraction [“Digital image processing of warm mix asphalt enriched with nanocolemanite and nanoulexite minerals”].
Answer: Thanks for the comments : In response to the concern raised, we have made revisions to the sentence in question, considering both the SEM images and relevant literature. The updated sentence now reads: "These images demonstrate that finely powdered microalgae, which underwent 6 hours of ball milling, was uniformly integrated into all MBBs, including those with 30% (Figure 5a), 40% (Figure 5b), 50% (Figure 5c), and 60% (Figure 5d) microalgae content. However, it was generally observed that microalgae particles exhibited irregular shapes and agglomeration due to the influence of van der Waals forces of attraction."We believe that these changes address the concerns raised by the reviewer and provide a more accurate description in line with both the SEM images and the cited literature.
In response to your feedback, the relevant modifications have been made, and you can review the details in Row 377-382 on page 11.
“These images demonstrate that finely powdered microalgae, which underwent 6 hours of ball milling, was uniformly integrated into all MBBs, including those with 30% (Figure 5a), 40% (Figure 5b), 50% (Figure 5c), and 60% (Figure 5d) microalgae content. However, it was generally observed that microalgae particles exhibited irregular shapes and agglomera-tion due to the influence of van der Waals forces of attraction.”
- After Fig.5, a general evaluation should be made as the last paragraph. In addition, the technological potential areas of use of the material obtained from this study should be given, and recommendations should be given to researchers who will study this subject in the future.
Answer: Thanks for your suggestion. Recently, there has been growing attention to environmentally friendly materials, especially in the context of ESG (Environmental, Social, and Governance) practices. Consequently, the redevelopment of biodegradable plastics has become increasingly vital. Our research is aimed at reducing the use of conventional petroleum-based plastics and replacing them with eco-friendly alternatives. While the ideal goal would be a 100% replacement of conventional plastics, practical and physical limitations exist. Therefore, it is crucial to make efforts to overcome these limitations from a long-term perspective.
In response to your feedback, the relevant modifications have been made, and you can review the details in Row 405-438 on page 13.
Reviewer 3 Report
Comments and Suggestions for Authors
The direction of this research is interesting, but overall it requires very serious revision.
1. I didn't find figure 1.
Methodological notes:
2. The filling level of milling container is not noted.
3. In the milling conditions, it is noted that the samples were milled at room temperature. However, the duration of the milling process was up to 12 hours, that inevitably leads to heating of the sample. It is not clear how this heating influences on the properties of the MBBs.
4. The moment of components mixing is not noted. It is not clear whether algae and EVA were milled together, or the components were milled separately and then mixed.
Notes on the discussion.
5. Discussion of data obtained using FTIR and DSK methods is clearly not enough. Probably much more information about the interaction of components with each other could be obtained. The conclusion that MBBs have nitrogen containing groups is obvious and does not have any novelty (FTIR).
6. 266 -268 lines: The authors noted that inflection points indicate the presence of complex components in algae, but there are no inflection points at the dependence for pure algae (red line in Figure 4 f). It can be seen that the inflection points on the TGA curve are the contribution of EVA.
7. The article contains a lot of data obtained with physicochemical methods such as FTIR and DSK, but these data are not yet related to the purpose of the study. Authors need to think how to use the obtained data and interpret them according to the purpose of the study. The purpose and conclusions should be modified according to the presented data after extensive discussion.
Author Response
- I didn't find figure 1.
Answer: We sincerely thank you for pointing out the mistake to our attention. Regrettably, we did make an error, and we have since revisited figures 1a, 1b, and 1c. We promptly recognized our mistake and made immediate corrections. We apologize once more for any oversight.
We have added a sentence based on your review (In row 202, page 6).
Methodological notes
- The filling level of milling container is not noted.
Answer: Thanks for the pointing out. The milling container has a volume of 125 mL, and we used 50g of balls and 20g of the material during the milling process.
We have added a sentence based on your review (In row 119-120, page 3).
“The milling container has a volume of 125 mL, and we used 50g of balls and 20g of the material during the milling process.”
- In the milling conditions, it is noted that the samples were milled at room temperature. However, the duration of the milling process was up to 12 hours, that inevitably leads to heating of the sample. It is not clear how this heating influences on the properties of the MBBs.
Answer: Thanks for the pointing out. The small volume of the milling jar and the size of the balls pose challenges in providing a detailed explanation of the heating effects. The experiment revealed a minimal temperature difference of only 2-3 degrees between the initial temperature measured before the experiment and the final temperature. Despite the potential heating during the 12-hour milling process, the observed temperature variation suggests that the impact on the overall temperature was relatively modest.
We have added a sentence based on your review (In row 106-108, page 3).
“After harvesting, the cells were subjected to controlled heat-spray drying. It is noteworthy that ball milling was exclusively applied to microalgae in this study.”
- The moment of components mixing is not noted. It is not clear whether algae and EVA were milled together, or the components were milled separately and then mixed.
Answer: We appreciate your observation. Indeed, in our process, the components were milled separately and then mixed. The algae underwent an individual milling process before being combined with EVA in the later stages of the experiment.
We have added a sentence based on your review (In row 121-126, page 3).
“The small volume of the milling jar and the size of the balls make it challenging to provide a detailed explanation of the heating effect. The experimental results showed a minimal temperature difference of only 2-3 degrees between the initial and final temperatures, indicating a negligible impact of potential heating during the 12-hour milling process.”
Notes on the discussion.
- Discussion of data obtained using FTIR and DSK methods is clearly not enough. Probably much more information about the interaction of components with each other could be obtained. The conclusion that MBBs have nitrogen containing groups is obvious and does not have any novelty (FTIR).
Answer: Thanks for the review. We acknowledge the need for a more in-depth discussion of FTIR and DSK data. In this study, we investigated the identification of similar functional groups between microalgae and EVA, which has not been extensively explored in previous research. This investigation is novel as it allowed us to overcome previous limitations, where only up to 20% mixing was achieved in prior studies. In contrast, our research successfully demonstrated effective blending exceeding 60%, showcasing the novelty and advancement in the field.
We have added a sentence based on your review (In row 52-65, page 2)
“Research on biomass-based biocomposite materials is a highly significant field. In previous research, a biocomposite was synthesized using PVA, incorporating a mixture of 20% LEA (microalgae) and 12% PD, resulting in a total composition of 68% PVA[PVA ì„ í–‰ì—°êµ¬]. In previous research established that the maximum synthesis ratio achievable was 20%. It was observed that mixing beyond this threshold, specifically exceeding 30%, posed limitations in the composite formation process.
This research explored avenues for the effective utilization of biomass. The focal point of our research revolved around determining the maximum extent to which microalgae content could be increased and how the physicochemical properties and characteristics changed concerning the mixing ratio. This emphasis on exploring the upper limits of microalgae content and dissecting the alterations in physicochemical properties based on mixing ratios set our study distinctly apart from prior research efforts. Furthermore, it played a pivotal role in identifying the optimal blending ratio, driving pioneering innovations in the realm of biocomposite material research.”
- 266 -268 lines: The authors noted that inflection points indicate the presence of complex components in algae, but there are no inflection points at the dependence for pure algae (red line in Figure 4 f). It can be seen that the inflection points on the TGA curve are the contribution of EVA.
Answer: Thanks for the pointing out. We apologize for any confusion caused. We initially stated 'negligible inflection' but have now revised it to 'no significant inflection' in response to their feedback. Our rationale for this adjustment is rooted in the fact that pure substances typically exhibit significant inflection points. However, since microalgae itself is not a pure substance, our results did not show any inflection points (i.e., none were observed).
We have added a sentence based on your review (In row 365-366, page 11).
“In contrast, the TGA curve of the microalgae did not exhibit a distinct weight loss (Figure 4e) or no significant inflection point (Figure 4e)”
- The article contains a lot of data obtained with physicochemical methods such as FTIR and DSK, but these data are not yet related to the purpose of the study. Authors need to think how to use the obtained data and interpret them according to the purpose of the study. The purpose and conclusions should be modified according to the presented data after extensive discussion.
Answer: Thanks for the review. The FT-IR analysis of the biocomposite material reveals that while there are discernible peaks corresponding to the microalgae component, the EVA peaks closely resemble those in the biocomposite. This similarity indicates that the primary component responsible for the observed peaks in the biocomposite material is EVA. Additionally, the presence of the microalgae peaks in the biocomposite suggests that the microalgae component blends well with EVA and does not significantly alter the material's properties.
We have added a sentence based on your review (In row 51-65, page 2 and row 426-441, page 13).
Based on the review comments :“Research on biomass-based biocomposite materials is a highly significant field. In previous research, a biocomposite was synthesized using PVA, incorporating a mixture of 20% LEA (microalgae) and 12% PD, resulting in a total composition of 68% PVA[PVA ì„ í–‰ì—°êµ¬]. In previous research established that the maximum synthesis ratio achievable was 20%. It was observed that mixing beyond this threshold, specifically exceeding 30%, posed limitations in the composite formation process.
This research explored avenues for the effective utilization of biomass. The focal point of our research revolved around determining the maximum extent to which microalgae content could be increased and how the physicochemical properties and characteristics changed concerning the mixing ratio. This emphasis on exploring the upper limits of microalgae content and dissecting the alterations in physicochemical properties based on mixing ratios set our study distinctly apart from prior research efforts. Furthermore, it played a pivotal role in identifying the optimal blending ratio, driving pioneering innovations in the realm of biocomposite material research.” and
“Successfully, MBBs were created through the blending of finely powdered microalgae with EVA. These blends accommodated microalgae contents of up to 60 wt%. The mechanical performance evaluation of the resulting MBBs revealed an impressive tensile strength of approximately 2.5 MPa, marking a significant achievement (equivalent to about 62.5% of the tensile strength of pure EVA). It is acknowledged that the performance of MBBs is inferior to pristine EVA. it is crucial to highlight the substantial influence of the ball-milling process. For instance, MBBs containing 60% untreated microalgae exhibited a tensile strength of only 0.7 MPa, approximately one-fourth that of MBBs with 60% treated microalgae. This increase in the mixing ratio set our research apart from previous studies. It played a crucial role in finding the optimal mixing ratio and spearheading innovative advancements in the field of biocomposite material research. This observation underscores an alternative approach for effective preconditioning of microalgae, engendering high-performance MBBs that align with economic viability. The combination of HS2 (Microalgae) and EVA in this biocomposite material holds significant environmental value and offers substantial potential as a futuristic material resource.” We have incorporated the changes into this paper.
Round 2
Reviewer 2 Report
Comments and Suggestions for Authors
The authors have mostly followed the reviewer's recommendations. Therefore, this manuscript is acceptable.
Comments on the Quality of English LanguageThe English language of the manuscript is sufficient.
Reviewer 3 Report
Comments and Suggestions for Authors
I express my gratitude to the authors for their attentive attitude to my comments